# Ozone as Modulator of Resorption and Inflammatory Response in Extruded Nucleus Pulposus Herniation. Revising Concepts

**DOI:** 10.3390/ijms22189946

**Published:** 2021-09-14

**Authors:** María de los Ángeles Erario, Eduardo Croce, Maria Teresita Moviglia Brandolino, Gustavo Moviglia, Aníbal M. Grangeat

**Affiliations:** 1Instituto Argentino de Ozonoterapia (IAOT), Buenos Aires C1425ASG, Argentina; angeleserario@iaot.com.ar (M.d.l.Á.E.); eacroce@iaot.com.ar (E.C.); 2Research Center for Tissue Engineering and Cell Therapy (CIITT), Civil Association for Research and Development of Advanced Therapies (ACIDTA), Buenos Aires C1425DKA, Argentina; teresitamoviglia@hotmail.com (M.T.M.B.); gmoviglia@gmail.com (G.M.)

**Keywords:** ozone therapy, disc herniation, intervertebral disc, nucleus pulposus, extrusion

## Abstract

Ozone therapy has been used to treat disc herniation for more than four decades. There are several papers describing results and mechanism of action. However, it is very important to define the characteristics of extruded disc herniation. Although ozone therapy showed excellent results in the majority of spinal diseases, it is not yet fully accepted within the medical community. Perhaps it is partly due to the fact that, sometimes, indications are not appropriately made. The objective of our work is to explain the mechanisms of action of ozone therapy on the extruded disc herniation. Indeed, these mechanisms are quite different from those exerted by ozone on the protruded disc herniation and on the degenerative disc disease because the inflammatory response is very different between the various cases. Extruded disc herniation occurs when the nucleus squeezes through a weakness or tear in the annulus. Host immune system considers the nucleus material to be a foreign invader, which triggers an immune response and inflammation. We think ozone therapy modulates this immune response, activating macrophages, which produce phagocytosis of extruded nucleus pulposus. Ozone would also facilitate the passage from the M1 to M2 phase of macrophages, going from an inflammatory phase to a reparative phase. Further studies are needed to verify the switch of macrophages.

## 1. Intervertebral Disc

The intervertebral disc (IVD) is placed between two cartilaginous endplates of adjacent vertebrae in the spine, providing mobility and support to the spine. IVD is made of three major components: an annulus fibrosus (AF), the nucleus pulposus (NP) and a cartilaginous endplate (CEP) [1,2,3].

The IVD is derived from embryonic structures: sclerotome and notochord. With the formation of vertebrae, sclerotome condenses around the notochord to form the vertebrae and the putative AF. At the same time, notochordal is contracted from the vertebral body and expands into the area of the future NP [4]. When the bony vertebra is formed, hyaline cartilage adjacent to the IVD is maintained and develops into the cartilage at the end plate. In the early stage of human life, the NP is populated by clusters of large, vacuolated notochordal cells and by small chondrocyte-like cells. However, by the second decade of life, the notochordal cells in the NP disappear, and the NP transitions from a notochordal structure to a tissue embedded with small chondrocyte-like cells. During this process, it is noteworthy that the NP tissue is sealed and isolated from the immune system ever since its formation at the beginning of IVD development [4,5,6,7].

Immune privilege organs are defined as regions in the body where foreign tissue grafts can survive and extend for indefinite periods of time, while similar implants placed at regular regions of the body are acutely rejected.

The existence of machinery that limits immunocytes and immune mediators entering the NP tissue in IVD has been suggested. This machinery, here, could be defined as a blood-NP barrier (BNB), which is a complex composition of physical and molecular factors. From an anatomical point of view, the BNB is a region that isolates the NP tissue from the host immune system. The AF and the CEP constitute a strong basement isolating the NP tissue from the host immune system. The intervertebral disc forms prior to the immune system [7].

Studies showed that Fas ligand (FasL), which is an apoptosis inducer and widely expressed in other immune privilege sites, exists in human NP tissues. It has been found that FasL could induce apoptosis of both vascular endothelial cells and immunocytes including macrophages and CD8+ lymphocytes. These studies indicate that FasL might act as a molecular barrier by eliminating blood vessel infiltration and immune cells recruitment [7,8,9,10,11,12,13]. The protective effect of notochordal cells in IVD and its suppressive impact on inflammation has been hypothesized. The AF, the CEP and molecular factors such as FasL establish a unique architecture for immune privilege [7,12,13].

It has been shown that fissure of AF is mechanically and chemically conducive to the ingrowth of blood vessels. The auto-immune response and downstream cascade reaction starts when the BNB is damaged. The radicular pain of a lumbar disc herniation results from the exposure of the NP and related auto-immune response. The auto-immune reaction could stimulate immunocytes and inflammatory cytokines infiltration, and these factors could in turn impact the IVD with harmful influence [7].

The normal IVD is considered an organ that is poorly innervated, supplied only by sensory and sympathetic perivascular nerve fibers. Most of the studies performed in different animal species, including humans, have demonstrated that nerve fibers in IVDs are found mostly in the periphery of the AF [14].

## 2. Low Back Pain as a Consequence of Disc Herniation

One of the most important clinical problems affecting humans worldwide is low back pain, with significant social and economic impact. Patients usually show back pain, radicular or sciatic symptoms, or both back pain and radicular pain caused by disc herniation 90% of the time. The reported incidence of lumbar disc herniation in the American population is 1–2% per year, and up to 70–85% of the general population has suffered at least one episode of low back pain in their lifetime. It is the most common cause of disability in North America [15].

Lumbar disc herniation (LDH) is a localized displacement of intervertebral disc tissue beyond the anatomical margins of the intervertebral disc space that can result in low back pain, radicular pain, motor weakness, numbness, and/or tingling in a myotomal and dermatomal distribution [16].

It is necessary to distinguish different pathological situations in the spine. It is not the same protruded disc herniation, extruded disc herniation, degenerative disc disease, etc. Therefore, we have reviewed the most recommended definitions about disc nomenclature. According to Fardon et al., protruded disc is one of the two subcategories of a “herniated disc” (the other being an “extruded disc”) in which disc tissue extends beyond the margin of the disc space involving less than 25% of the circumference of the disc margin as viewed in the axial plane. While sometimes used as a general term in the way herniation is defined, the use of the term “protrusion” is best reserved for subcategorization of herniation meeting the above criteria. An extruded disc is a herniated disc in which, in at least one plane, any distance between the edges of the disc material beyond the disc space is greater than the distance between the edges of the base of the disc material beyond the disc space in the same plane, or when no continuity exists between the disc material beyond the disc space and that within the disc space. The definition is consistent with the common image of extrusion as an expulsion of material from a container through and beyond an aperture. Displacement of disc material beyond the outer annulus with any distance between its edges greater than the distance between the edges of the base distinguishes extrusion from protrusion. Characteristics of protrusion and extrusion may coexist, in which case the disc should be subcategorized as extruded. Extruded discs in which all continuity with the disc of origin is lost may be further characterized as “sequestrated”. Disc material displaced away from the site of extrusion may be characterized as “migrated” [17].

## 3. Disc Herniation: Immune Phenomenon

The disc may herniate due to a disruption of the annulus fibrosus or due to its attachments to the adjacent endplate.

When rupture of the AF occurs with direct exposure of the nucleus pulposus (NP) to the host’s immune system, a primary inflammatory reaction starts. The autoimmunity reaction caused by the prominent NP tissue is one of the most important mechanisms for reabsorption after the disc herniation [18].

The inflammatory and immune activation profile exhibited by a herniated disc is different in extrusion than protrusion. It is proven that an intervertebral disc has inflammatory-like cells capable of releasing inflammatory mediators that can help recruit cells from the immune system. Particularly, the monocyte chemoattractant protein (MCP1) a CC chemokine and the IL-8, whose main functions are the chemotaxis of macrophages and angiogenesis. Extruded disc has infiltration of macrophages, TNFα, IL-1β, IL-6, IL-8, prostaglandin E2 (PGE2), and nitric oxide (NO). In herniated disc fragments, there are greater expression of IL-12 and interferon-γ (IFN γ) compared to bulging discs that remain contained within the disc space by an intact annulus fibrosus. Most of these cytokines are macrophage products and can promote lymphocyte activation and differentiation while also recruiting additional macrophages and activating phagocytosis and secretion of proteolytic enzymes. Moreover, Th1 lymphocytes produce IFN γ, which recruits and activates more macrophages. There is a great presence of macrophages and its products in extruded discs—elevated expression of CD4 lymphocytes, too. This provides support to the fact that NP tissue contact with the systemic circulation in disc herniation leads to lymphocyte activation with secretion of IFN γ and consequent macrophage recruitment, with extruded nucleus pulposus resorption and inflammatory peripheral radiculopathy manifesting as sciatica [19,20,21,22,23,24].

The elevated expression of IFN γ in herniated discs may represent a specific immune response against herniated NP tissue [23].

After exposure of the autologous NP, lymphocytes accumulate in the regional nodes. In addition, after the rupture of the AF, lymphocytes accumulate in the intervertebral disc (IVD), and there is a deposit of complement and immunoglobulins in the herniated tissue [25,26,27,28].

These findings illustrate a pattern of immune activation by herniated disc tissue involving macrophage infiltration and activation [23].

Studies have shown the inflammatory processes and elevated production of prostaglandins, leukotrienes, and thromboxane in herniated disc tissues [21,29].

Extruded disc herniation has elevated production of IL-6, IL-12, IFN γ, and presence of CD68 macrophages. This suggests a pattern of TH1 lymphocyte activation.

Tissue-level consequences of this recruitment could include desirable NP tissue resorption and undesirable inflammatory peripheral neuropathy manifesting as sciatica [23].

## 4. Natural Evolution of Extruded Disc Herniation

Extruded disc herniation resorption occurs via an inflammatory reaction with macrophages as the predominant cellular population. Migrating and extruding herniation types have a higher tendency to decrease in size in follow-up studies. The vast majority of extruded disc herniation had resolved spontaneously within one year. Clinical symptom alleviation occurs concordantly with faster resorption rate. In most cases, the natural course of extruded disc herniation involves its reduction in size over time [30].

The macrophage infiltration is believed to exacerbate pain symptoms through secretion of pro-inflammatory cytokines such as IL-6, IL8, and TNF-alfa. In contrast, macrophage infiltration may also have a positive effect on symptoms through inducing a phagocytic resorption process, mediated by anti-inflammatory cytokines such as IL-4 and IL-10.

Based off a one year follow up in patients with sciatica, it is thought that macrophage infiltration is associated with an extruded type of disc herniation as well as the extent of reduction. Macrophages have an active role in resorption of extruded disc herniation. The degree of macrophage infiltration was higher in extrusion in comparison to bulging (protrusion) of the disc. Results were comparable in patients with and without Modic changes [24].

Currently, the literature distinguishes M1 and M2 macrophages. M1 produces pro-inflammatory cytokines and products such as IL-1, IL-1α, IL-1β, IL-6, IL-8, IL-12, IL-18, IL-23 IL-27, TNF-α, and Bone Morphogenic Protein 2 (BMP-2), and its expression profile is associated with exacerbation of pain symptoms. M2 macrophage excretes anti-inflammatory cytokines such as IL-1Ra, IL-10, IL-4 and transforming growth factor-beta (TGF-β), which are involved in multiple functions such as tissue repair and remodeling. M2 is believed to alleviate pain symptoms through resorption of herniated disc material. During most inflammation processes, M1 or M2 macrophages occur sequentially [31,32,33,34]. Cytokines excreted during the process of disc herniation in sciatica seem to have a different effect on pain symptoms. Pro-inflammatory cytokines worsen pain symptoms, while anti-inflammatory cytokines alleviate pain symptoms [34] (Figure 1).

The painful radiculopathy, often with sensory and/or motor deficit, that accompanies extruded disc herniation originates from the immune system’s reaction to the presence of a foreign body (nucleus pulposus): phase M1.

All these mechanisms allow the resorption and remission of the herniated portion of the nucleus pulposus (NP), which is outside the intervertebral space, by the physiological mechanisms of lysis and phagocytosis. It occurs because this portion of the NP is exposed to the direct action of the immune system, while the rest of the NP within the intervertebral space is protected from the invasion of immune cells, thanks to Fas ligand. Large aggrecans molecules prevent cytokines from entering too.

It has been pointed out that for a clinical improvement, a regression of at least 20% of the herniated disc must occur, although the translation of this clinical improvement is established later in magnetic resonance images (MRI) [35].

## 5. Ozone Therapy and Its Implication on Extruded Disc Herniation Resorption. Macrophages: The Master Key

Ozone therapy has been widely used in everyday clinical practice over the last few years, leading to significant clinical results in the treatment of herniated discs and pain management [36].

Ozone (O_3_) is a highly unstable gas that rapidly decomposes to oxygen, and it is clinically used as an oxygen-ozone gas mixture at low concentrations for multiple therapeutic purposes. O_3_ is a natural gas forming from dioxygen (O_2_) by the action of ultraviolet light and electrical discharges in the atmosphere; however, O_3_ is highly unstable, rapidly breaking down to its diatomic allotrope. For this reason, it occurs in very low amounts in the atmosphere compared to O_2_. O_3_ is known as a strong oxidant being at the same time a dangerous respiratory hazard and pollutant, contributing to several diseases; it is also a powerful oxidizing agent with manifold industrial applications [37].

Ozone has a paradoxical effect because it can be a very strong and dangerous oxidant, while, in appropriate concentrations, it acts as a real beneficial drug. This is because most of the medical ozone dose is almost instantly quenched by the potent antioxidant capacity of blood due to a number of hydrophilic, lipophilic compounds and a variety of antioxidant enzymes. This explains why ozone can be used safely in medicine [38].

When ozone gets in contact with human fluids and tissues, it reacts with polyunsaturated fatty acids (PUFA), creating hydrogen peroxide (H_2_O_2_) and a mixture of lipid ozonation products (LOP), mainly composed by 4-HNE (from omega-6 PUFA) and 4-HHE (trans-4 hydroxy-2-hexenal from omega-3 PUFA). H_2_O_2_ is an important reactive oxygen species (ROS), and it acts as a messenger with extremely short lifetime (a few seconds) [39,40,41,42,43,44].

The moderate oxidative stress caused by ROS is counteracted by endogenous radical scavengers, such as superoxide dismutase, glutathione peroxidase, catalase, and NADPH quinone-oxidoreductase. It has been shown that small and repeated oxidative stresses could induce the activation of transcriptional factor mediating nuclear factor erythoid 2 related factor 2 (Nrf2), a domain involved in the transcription of antioxidant response elements (ARE). Therefore, repeated mild oxidative stress produced by ozone could induce upregulation of Nrf2 and transcription of AREs, responding better to pathological oxidative stress [44,45,46].

Low doses of ozone preconditioning can activate the Nrf2 pathway and the promotion of the feedback mechanism that induces the synthesis of proteins, which collectively favors cell survival [47].

Evidence suggests that the therapeutic effects of ozone treatment activate the Nrf2 pathways. Nrf2 is a master regulator of the genes that protect cells from the effects of endogenous and exogenous insults. In particular, Nrf2 regulates the expression of genes under the control of ARE enhancers. Under basal conditions, Nrf2 is sequestered in the cytoplasm by its specific inhibitor Keap1 (Kelch-like ECH associated protein). Under specific stimuli, Nrf2 dissociates from Keap1, translocates into the nucleus and trans-activates ARE-driven genes [48].

Gallie et al. recently showed that Nrf2 activation and upregulation of the stress responsive genes (HO-1, ERCC4, CDKN1A) may be observed when cells are ozonized in culture. Low ozone concentration stimulates cell protective pathways and nuclear transcription via the activation of an antioxidant response through induction of an oxidative “eustress” able to stimulate cell defense pathways without causing deleterious effects [48].

There are discrepancies in the medical community regarding the optimal concentrations used to obtain the best results. Taking into account the works of Renate Viebhan and other authors, we believe that different concentrations produce different metabolic effects. Viebhan has described that ozonated autohemotransfusion (AHT) at 20, 25 and 30 µg/mL stimulates AREs and antioxidant response, but this has a different repercussion on the kidneys. In an in vitro model, Costanzo et al. have demonstrated that the effects of ozone administration are dependent on gas concentration, and that the nucleus and the cytoplasmic organelles may be differently affected. They studied the direct effects of mild ozonization on some cellular mechanisms. They proved that ozone at 10 μg/mL induces positive and long-lasting cellular responses in cytoskeletal organization and mitochondrial activation, as well as in nuclear transcription. On the other hand, 1 μg O_3_/mL-treatment seems to represent a stimulus able to activate some transient responses [49,50]. Above the immune modulatory concentrations, ozone can generate toxic effects.

The low-dose ozone concept with its moderate oxidative stress represents an ideal hormesis strategy. Dose-response and concentration-effect relationships in the context of specific applications allow one to fix concentration ranges with therapeutic benefit. In its pharmacological effect, medical ozone follows the principle of hormesis: low concentrations (or doses) show high efficacy, which decreases with increasing concentration [50].

Intradiscal oxygen-ozone (O_2_-O_3_) injection has been tested in large clinical studies, which have shown a positive outcome in 70–80% of patients. In a randomized controlled study to assess the effectiveness of intraforaminal injection of O_2_-O_3_ versus steroids, it has been demonstrated that O_2_-O_3_ injection is more effective than steroids [51].

Wang et al. showed that intrathecal injection of low-concentration O_3_ reduces the overexpression of TNF-α, IL–6, and IL-1β and also alleviates mechanical allodynia in rats with chronic radiculitis [52].

Ozone provides rapid and long-lasting relief of pain through minimal invasive techniques, without any adverse effects. Ozcan et al. suggest 10 mL of mixture of O_2_-O_3_ at 25 μg/mL concentration for intradiscal injection, and they prefer higher volume to avoid potential triggering of pain by high concentrations [53].

After having 19 years of experience doing intradiscal injection in more than 3500 patients, we fully agree with this concept. We think that the most adequate concentration for intradiscal injection to treat extruded disc herniation is between 20 and 30 μg/mL. In our experience, better results were obtained with 25 μg/mL because we are looking for an immune modulated effect, and this is because the effect is the immune modulated effect that resolves disc herniation, and it is not necessary to use corticosteroids.

In all these 19 years, we have been treating extruded disc herniation with ozone therapy and rehabilitation (physiotherapy) without corticosteroids.

It has been described that intradiscal ozone injection alone was sufficient to treat low back and leg pain caused by lumbar disc herniation, and that periforaminal steroid injection does not provide additional benefit [54,55].

Paravertebral intramuscular O_2_O_3_ injections have proved to provide pain relief and decreased medication intake in people with low back pain caused by herniated discs. The mechanism of action is based on the biochemical properties of O_3_: immunomodulatory action, and analgesic and anti-inflammatory effects [56].

Substantial work has shown ozone therapy treatment of contained herniated intervertebral discs to be safe and effective in relieving pain. Although the precise mechanism of action of ozone remains undetermined, prior studies suggest it may interact with disc proteoglycans and/or modulate inflammation [57,58,59].

Murphy et al. indicate as main mechanisms of action of ozone intradiscal injection the fragmentation of glycosaminoglycans with subsequent dehydration of the intervertebral disc and inflammatory response through the cytokine cascade. In this model, they are referring to contained disc herniation too.

The same authors created a mathematical disc model, which predicts that therapeutic doses of ozone produce a disc volume reduction in 6%, resulting in a height reduction of 0.025 cm and a 9.84% pressure reduction [59].

All these papers explain the mechanism of action of ozone in the contained disc herniation. This explanation is through the reduction in the volume of the herniated disc due to dehydration, with the consequent decrease in intradiscal pressure. This would decompress the root and relieve pain, but it applies to contained herniated disc, not to extruded disc herniation.

In recent decades, the importance of the inflammatory mechanism in the etiology of the painful radiculopathy that accompanies a herniated disc has been described.

We have been observing for the last 15 years that there is mechanism of action of ozone other than intervertebral disc dehydration on extruded disc herniation. We have been watching that on patient’s MRI checkups after a few months of ozone treatment, intervertebral discs treated with ozone have revitalized nucleus pulposus (NP). Additionally, Buric’s observations are in coincidence with ours. He observed that, during long-term follow-up of patients, MRI imaging showed that treated discs stayed hydrated and did not become a low signal [60].

This observation contradicts the mechanism of disc dehydration proposed for ozone therapy. In 2012, we wrote our first preliminary observation: ozone therapy modulates oxidative stress that takes place in the disc-radicular conflict and modulates the immune system, which is the one that resolves the extruded portion of NP through phagocytosis and lysis by activated macrophages. This is more different in protruded (or contained) discs than in extruded disc herniation [61,62].

According to the response to treatment with ozone given by the predominant tissue in the herniated material, three types of herniated disc can be distinguished: extruded disc herniation (NP), protruded disc herniation (AF), protruded disc herniation with tear in the AF.

## 6. Ozone Therapy as Immunomodulator in Extruded Disc Herniation

There is plenty of evidence about immunomodulatory effect of ozone. Several hypotheses have been proposed to clarify the mechanisms underpinning the antioxidant, antalgic, anti-inflammatory and immunomodulatory actions of medical O_2_O_3_ mixture. It has been said above that Nrf2 play an important role in the intracellular signaling pathways of inflammation. The activation of Nrf2-antioxidant signaling might attenuate NF-kB, a key regulator of inflammation [44].

It has been described that ozone therapy decreased phagocytosis of polymorphonuclear (PMN) in healthy cows but increased in cows with mastitis and fever milk. Therefore, ozone activates macrophages phagocytosis when it is necessary [63].

An increased phagocytic activity of PMNs and relative number of phagocytic neutrophils, as well as absorption and digestion of the bacteria used in an experimental model [64], have been demonstrated.

Low doses of O_3_ might also have a role in the regulation of prostaglandin synthesis, the release of bradykinin, and in increasing secretions of macrophages and leukocytes [44]. The ozone immune modulation effect is based on the oxidative metabolism and on the immune system. It was demonstrated that ozone was able to modulate the phagocytic cells in peripheral blood and the mechanisms on how messengers can activate immunological response leading to the therapeutic biological effects. Furthermore, it was demonstrated that there is a range of ozone concentrations where we can obtain the highest positive results, while lower doses are ineffective and higher doses can produce lower effects. Accordingly, ozone, in a dose-dependent behavior, may stimulate the phagocytic function of the peripheral blood cells [65]. The immunological action of ozone on the blood is mainly directed at monocytes and T lymphocytes, which once induced, release small amounts of cytokines. During ozone therapy, the release of antagonists of cytokines or cytokines—such as IL-10, IL-4 and TGF-β which are capable of suppressing autoreactive cytotoxicity, can increase; therefore, the induction of cytokines would not exceed levels beyond those necessary once counter-regulatory elements are activated. In a rat model of aseptic inflammation, the phagocytic activity of ozone has been studied. The results revealed that a rise in phagocytosis according to time treatment increased. These modulating characteristics of ozone therapy are seen in some paradoxical findings. Positive results have been achieved both in patients who have an exacerbated immune response, as well as in patients with autoimmune diseases or with immunological deficits [66].

According to the literature consulted, it is clear that when the nucleus pulposus (NP) is exposed to the immune system (extruded disc herniation), an antibody–antigen reaction takes place, and then an inflammatory cascade occurs.

Reactions that occur inside and outside the disc space are different. The explained mechanisms of resorption and remission of the extruded NP by lysis and phagocytosis are only for extruded herniation because the immune privileged NP is exposed to the immune system. The disc that remains within the disc space is protected by the Fas ligand. In protruded disc herniation, the inflammatory reaction and the mechanism of action of ozone are different.

## 7. What Is the Role of Ozone Therapy in this Pathology?

The efficacy of ozone therapy in the painful pathology of the spine has been widely demonstrated. However, we think that the described mechanisms are attributed to the contained herniated disc.

We know that the therapeutic doses of ozone decrease the activation of NFkB and therefore the action of pro-inflammatory cytokines, as well as the metabolism of arachidonic acid. Ozone therapy reduces the auto immune inflammatory reaction and the pain of the radiculopathy, which accompanies the primary disease: the exposure of the nucleus pulposus to the immune system. Ozone therapy induces resorption of the herniated portion of the nucleus pulposus as an autoimmune physiological phenomenon.

## 8. Discussion

In the extruded disc herniation when the nucleus pulposus (NP) is exposed to the immune system, an antibody–antigen reaction takes place, and then an inflammatory response and inflammatory mediators cascade occur. This autoimmune mechanism produces the resorption of the herniated portion. The process is slow and painful, and it never consolidates. Therapeutic doses of ozone produce activation of Nrf2-antioxidant signaling consequently attenuating activation of NF-kB, a key regulator of inflammation, therefore reducing pain. In these concentrations, ozone stimulates macrophages phagocytosis of the herniated portion and its complete resorption, preserving the NP revitalized within the intervertebral space, improving the quality of the final scar formation. That is why we believe that ozone would also facilitate the passage from the M1 to M2 phase of macrophages, going from an inflammatory phase to a reparative phase. Further studies are needed to verify the switch of macrophages.

Regarding what it has been said, it can be stated categorically that ozone therapy is a therapeutic modality that solves the etiology of the extruded disc herniation; therefore, it should be considered as the treatment of choice for this pathology. Moreover, it is necessary to clarify that, due to what it has been stated, ozone therapy must be excluded from the concept of ‘pain treatment’ in this pathology. The pain goes into remission because the pathology that causes it is solved.

Likewise, we suggest that the term ‘discolysis’ should no longer be used, and it should be replaced with ‘ozone therapy’. Discolysis is a term very often used in medical gatherings as well as in our colleagues’ scientific works because it refers to a lysis in the IVD, which is not so; actually, it has been observed, ten years after performing the treatment with ozone, that the NP remains revitalized within the intervertebral space [60].

## 9. Conclusions

Ozone therapy does not produce discolysis in extruded disc herniation because it preserves the nucleus pulposus revitalized within the intervertebral space. Ozone, in adequate concentrations, modulates the immune system that solves the inflammation, the pain, and reabsorbs the extruded portion of the nucleus pulposus. Because of the usage of ozone therapy, extruded disc herniation is solved without surgery; ozone therapy reduces the recovery time of neurological function, and it obtains a better final recovery quality.

## Figures and Tables

**Figure 1 ijms-22-09946-f001:**
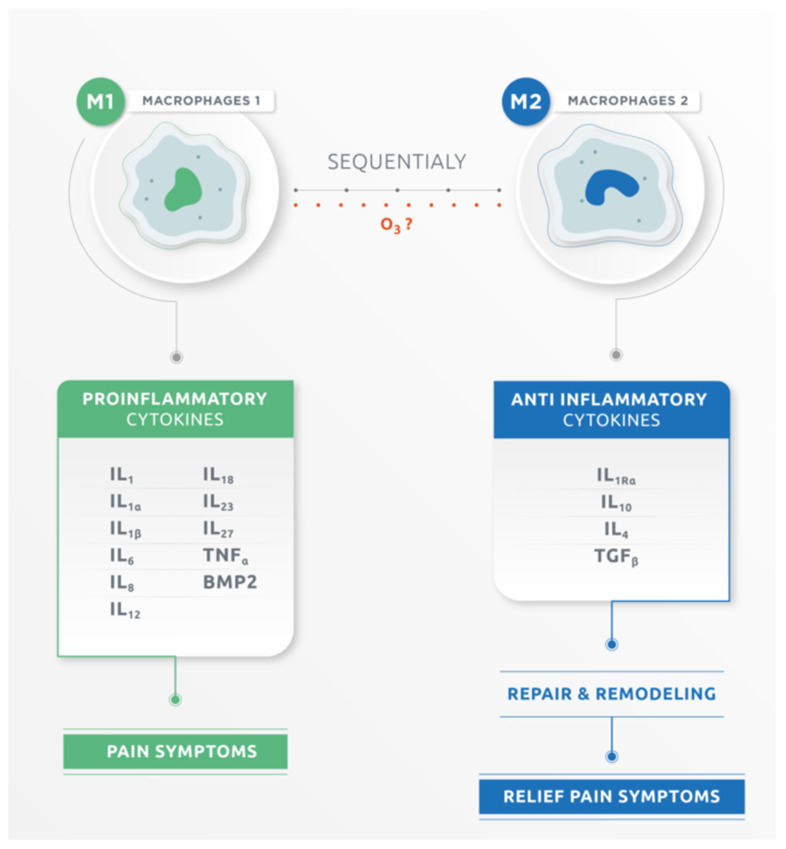
Macrophage M1 and macrophage M2. Macrophage M1 produces pro-inflammatory cytokines while macrophages M2 produces anti-inflammatory cytokines.

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
