# Peer review of "Ozone as Modulator of Resorption and Inflammatory Response in Extruded Nucleus Pulposus Herniation. Revising Concepts"

_ijms, 2021, doi:10.3390/ijms22189946_

Round 1

Reviewer 1 Report

In my opinion this is an article that fully describes the action of ozone therapy in the field of back pain. For many years, ozone treatment has been one of the most effective and least risky therapeutic solutions. I believe that this is the first work that attempts to explain the mechanisms underlying the effects obtained by many specialists in the sector from a pharmacological and biochemical point of view. I suggest to accept after minor revision

Author Response

Thank you very much for the comments made to our work. We have taken into account each of the suggestions made and we have corrected as follows.

  1. We removed therapy in the title, as suggested

Pag 1 Row 13: We reformulated the sentence highlighted in the abstract 

Pag 1 Row 15-17: We reformulated the redaction of the objective of our work presented in the abstract.

Page 8 Row 318. We corrected reference number 

Pag 9 Row 329: We corrected the author's last name.

We are very grateful for your kind comments and your favorable opinion of our work.

Reviewer 2 Report

Thank you for the opportunity to review your manuscript “Ozone therapy as modulator of resorption and inflammatory response extruded nucleus pulposus herniation. Revising concepts”

While the purpose of this review paper is to review the mechanism of treatment in patients with the extruded disc herniation, the manuscript is largely discussing about what has been already known (herniated disc). Although the topic is clinically relevant and the context is also educational, it feels like reading a textbook.

Generally, the manuscript is poorly written. For example, “Intervertebral disc” has 19 paragraphs and some of them only have one sentence. The anatomy and therapeutic concepts should be shortened. I also see other paragraphs that have a single sentence, which looks unbalanced and disorganised.

I see the first “Ozon therapy” in line 227. Is the “intervertebral disc section” necessary? I read the section as the introduction of the study.

As a review paper, the current form of this manuscript is structurally incorrect. For my experience, a review paper has several steps of methodology (search terms and criteria, evaluation tools, statistical analyses), which is missing. I will defer to the editor for this issue since I am not familiar with this type of review paper.

The section title in line 447 seems to be the results of this study. Even through thousands of words were written prior to line 447, the results are unclear and not scientifically supported; thus, unable to explain the mechanism of the therapy.

Discussion section is redundant to previous sections.

Conclusions do not reflect the purpose of the study. In other words, the conclusion sentences do not explain the mechanism of the therapy.

Minor corrections

Abstract: Do not mention protruded disc herniation and degenerative disc disease (throughout the manuscript) if these are not related to the main purpose of the study.

Figures: Illustrations are not clear. Also, figure 1 needs  captions.

Author Response

1) As you well observed, we had developed the intervertebral disc section very extensively. Following your suggestion, we have summarized it.

We have removed the paragraphs that did not have a substantial meaning for the purpose of the work. However, we conserved those that describe certain characteristics of the intervertebral disc that are important to understand the mechanisms of action of ozone in extruded disc herniation.

We are aware of its length, but in our 20 years of experience we have read many papers where the authors use the term herniated disc in a general way without distinguishing the different types of herniated discs, which leads to great confusion in the indication and treatments.

Each type of herniated disc generates by itself its own inflammatory response in the host, so the mechanism of action of ozone is different in each one. In this work we refer only to the extruded disc herniation, the immune and inflammatory response generated in the host and how ozone (immunomodulator) acts on it.

2) We have revised the paragraph layout based on your suggestion.

3) We reviewed the Discussion and Conclusions section following your suggestions.

4) In the abstract, as you suggested, we highlighted the extruded disc herniation, removing in one of the sentences the degenerative disc disease and protruding disc herniation (we only left it in a sentence at the end).

5) We put a subtitle to Figure (Graphical Abstract).

Round 2

Reviewer 1 Report

The manuscript describes for the first time the real mechanism underlying the ozone action on disc herniation. I recommend it for publication in this form.

Reviewer 2 Report

It has undergone one cycle of revision. In the revision report, the authors reported that they have taken into account each of the suggestions. I disagree with the authors’ statement that the yellow highlights (I assume that these are revised parts) are deceptive.

For example, the first sentence in the introduction “The intervertebral disc … … a cartilaginous endplate (CEP)” has not been revised at all but the references have only been added. In this case, the references [1-3] should be highlighted. Second, the next paragraph “The IVD … … the future NP.” has been relocated (originally in line 87-90) and a reference has been added. Same way that a reference has only been added for the third highlighted part “The normal IVD … … the periphery of the AF.” I do not consider these changes as “revision”.

The manuscript has not been improved and I do not recommend publication.